# Chitosan-Polyphenol Conjugates for Human Health

**DOI:** 10.3390/life12111768

**Published:** 2022-11-02

**Authors:** Ananya Pattnaik, Sanghamitra Pati, Sangram Keshari Samal

**Affiliations:** 1Laboratory of Biomaterials and Regenerative Medicine for Advanced Therapies, ICMR-Regional Medical Research Center, Bhubaneswar 751023, Odisha, India; 2KSBT, Kalinga Institute of Industrial Technology, Bhubaneswar 751024, Odisha, India

**Keywords:** polyphenol, chitosan, antioxidant, oxidative stress, human health

## Abstract

Human health deteriorates due to the generation and accumulation of free radicals that induce oxidative stress, damaging proteins, lipids, and nucleic acids; this has become the leading cause of many deadly diseases such as cardiovascular, cancer, neurodegenerative, diabetes, and inflammation. Naturally occurring polyphenols have tremendous therapeutic potential, but their short biological half-life and rapid metabolism limit their use. Recent advancements in polymer science have provided numerous varieties of natural and synthetic polymers. Chitosan is widely used due to its biomimetic properties which include biodegradability, biocompatibility, inherent antimicrobial activity, and antioxidant properties. However, due to low solubility in water and the non-availability of the H-atom donor, the practical use of chitosan as an antioxidant is limited. Therefore, chitosan has been conjugated with polyphenols to overcome the limitations of both chitosan and polyphenol, along with increasing the potential synergistic effects of their combination for therapeutic applications. Though many methods have been evolved to conjugate chitosan with polyphenol through activated ester-modification, enzyme-mediated, and free radical induced are the most widely used strategies. The therapeutic efficiency of chitosan-polyphenol conjugates has been investigated for various disease treatments caused by ROS that have shown favorable outcomes and tremendous results. Hence, the present review focuses on the recent advancement of different strategies of chitosan-polyphenol conjugate formation with their advantages and limitations. Furthermore, the therapeutic applicability of the combinatorial efficiency of chitosan-based conjugates formed using Gallic Acid, Curcumin, Catechin, and Quercetin in human health has been described in detail.

## 1. Introduction

Reactive Oxygen Species (ROS) play an important role as intermediates in various metabolic pathways, but the accumulation of ROS induces Oxidative Stress (OS), damaging proteins, lipids, and nucleic acids. This has become the major cause of several deadly diseases. Epidemiological evidence indicates that the risk of premature mortality from major clinical conditions like cardiovascular disease (CVD), neurodegenerative disease, and cancers can be decreased by a high dietary intake of polyphenols [1,2,3,4,5,6]. Polyphenols are naturally occurring secondary metabolite micronutrients in plants, which contain hydroxyl groups in their aromatic ring [7,8,9,10]. The pictorial representation of different classes and subclasses of polyphenols, their chemical structures, and their sources are shown in Figure 1 [11]. Polyphenols form a natural defense system for plants, protecting them from UV radiation and pathogenic invasion, and are also responsible for oxidative stability and organoleptic properties [12,13,14,15,16,17].

Polyphenols attract immense attention for their potential nutraceutical and pharmaceutical impacts on human health, due to their inherent biological properties such as antioxidant, anti-allergic, antibacterial, anti-inflammatory, antitumor, anti-diabetic, and antiviral [18,19,20,21,22,23,24,25,26,27,28,29,30]. They also keep blood vessels healthy and flexible by managing blood pressure levels that is helpful for good blood circulation [31,32,33]. Hence, health professionals, researchers, and consumers are interested in natural antioxidants based on polyphenols to be incorporated into various therapeutic products and medicines [34,35,36,37,38]. Polyphenols such as curcumin [39,40,41], resveratrol [42], quercetin [43], and catechins [44] show protective abilities against neurodegenerative diseases (Alzheimer’s-like diseases and dementia) through their antioxidant, immunomodulatory, and scavenging properties. They inhibit the neurotoxic effects of accumulated beta-amyloid protein, which is a major protein linked to the cause of Alzheimer’s disease. Moreover, the chelating capacities of some polyphenols like curcumin, epigallocatechin gallate (EGCG), myricetin etc., with iron also prevent neurotoxicity [45]. They are also known to modulate inflammatory responses through the mitigation of cytokine pathways and hence prevent systemic and/or localized inflammation [46]. In the case of CVD, flavonoids and resveratrol are well known polyphenols that block cholesterol oxidation to reduce the risk of disease [47,48]. Anthocyanins are the most substantiated polyphenol that helps to regulate glycemic levels through the protection of β-cells from glucose toxicity; their their antioxidant and anti-inflammatory effects that leads to prevent and maintain type 2 diabetes [49,50]. Some polyphenols possess anti-obesogenic effects that help in weight loss and health maintenance by adipocyte oxidation, inhibition of lipogenesis, reduction in inflammation, and increase in energy expenditure [51]. The therapeutic actions of phenolic compounds for different diseases is pictorially represented in Figure 2 [52]. The ability of polyphenols to donate an electron/hydroxyl ion that binds and neutralizes the free radicals formed in the body [53]. This property also helps them bind to the biomacromolecules to form stable chemical compounds and reduces the ability of the complex to react further [54]. Polyphenols are also known to reduce OS by producing hydrogen peroxide. However, one of the negative impacts of polyphenols includes complex formation with biological elements inside the body. This results in a pro-oxidant nature under certain circumstances and their reduced efficiency due to various factors, i.e., pH of body fluid, presence of enzymes, and limited solubility [55]. Polyphenols are absorbed by the cell membrane, interact with the enzymes, and can chelate with metals, which leads to the mechanism of their antimicrobial nature [56,57]. However, due to short biological half-life, rapid metabolism, low water solubility and bioavailability, limits their success in clinic [58,59]. The absorption of polyphenols is dependent on its chemical structure and molecular weight by cells/tissues which also limits their oral administration. In addition, gastrointestinal enzymes, nutrients, microbiota, and other factors decrease their potential health benefits [60]. Moreover, environmental factors such as heat, moisture, oxygen, and light irradiation also affect the topical use of polyphenols due to degradation [61,62]. Therefore, many pharmaceutical and biomedical industries are hesitant to use polyphenol based compounds for translational medicine. All these limitations make it difficult for the systemic therapies of polyphenols [24,63,64,65,66,67,68]. Over the decades, continuous research is being made to overcome the deficiencies of polyphenols. Some of the well-used approaches include hydrogels, simple emulsions, nano-formulations, lipid encapsulation, liposomes etc. [24,69,70,71,72] In recent years, the conjugation of polyphenols with natural and/or synthetic polymers to enhance the therapeutic efficacy without losing any bioactive properties has been a spot of continuous attraction [72,73]. A widely studied natural cationic polymer, chitosan acts as an excellent delivery agent and is also being used as biomolecule conjugate for regenerative medicine along with inherent antibacterial, antioxidant, antitumor, and anti-inflammatory properties [74,75,76,77,78,79,80,81]. The radical scavenging nature of chitosan is due to active hydroxyl and amino groups that react with free radicals [82]. Several researchers have proposed that the binding of hydroxyl groups of chitosan to the free radicals forms a macromolecular stable compound that helps to reduce OS in the human body; whereas, some others have proposed that amino groups are react with free radicals [83]. Moreover, chitosan is also a potential metal chelator due to the presence of amino and hydroxyl groups which makes it easier for modification with other biomolecules [84]. However, low water solubility and poor availability of H-atom donor limit the practical use of chitosan as an antioxidant. Hence, grafting hydrophilic functional groups can increase solubility; providing H-atom donor can improvethe efficiency and applicability of chitosan [85]. Therefore, the conjugation of chitosan and polyphenol has become the most potentially beneficial agent to complement each other and enhance their therapeutic efficacy in human health. The flow diagram of the limitations and advantages of chitosan, polyphenols, and chitosan-polyphenol conjugates is given in Figure 3. Conjugation of polyphenols onto chitosan prevents the degradation of polyphenols in a biological environment (from light, pH, temperature, oxygen, enzymes, etc.) of the human body.

Many of the synthesis strategies to couple chitosan and polyphenol have been intensively investigated, and have been explored from chemical modifications, such as ester modification, enzyme-mediated, and free radical induced processes [86]. Several studies have been performed to evaluate the therapeutic efficiency of chitosan-polyphenol conjugates for various diseases [87,88,89,90,91]. However, not a single review has been made that only concentrates on chitosan-polyphenol conjugates for different disease treatments for human health. In this review, different recently used strategies of chitosan-polyphenol conjugate formation, with their advantages and limitations, have been described. Furthermore, the applicability of the combinatorial efficiency of both chitosan and polyphenol for therapeutic application in human health has been described in detail.

## 2. Chitosan-Polyphenol Conjugates

Chitosan-polyphenol conjugates impart properties of both components with high efficiency; however, both chitosan and polyphenol possess some disadvantages, such as poor solubility, dietary metabolism, and poor availability of H-atom donor etc., when used alone [85]. Conjugation of both has been able to overcome the limitations and shown a synergistic effect with tremendous potential as an antioxidant for various ROS-induced diseases [86]. Several strategies have been used for the conjugation, but chemical (activated ester-mediated modification), enzyme-mediated, and free radical induced are recently developed and widely used methods [86]. Activated ester modification uses 1-ethyl-3-(3-dimethylaminopropyl) carbodiimide (EDC), which is a carboxylic activating agent and act as a cross-linker to link chitosan and the polyphenol. EDC generates an O-acylisourea intermediate when added to polyphenols by activating the carboxyl group; it then couples with primary amine group of chitosan through amide bonds (Figure 4A). However, several studies have reported that there may be a possibility of hydrolysis and carboxyl group regeneration from the intermediate [92] and hence, introduction of N-hydroxysuccinimide (NHS) after the formation of O-acylisourea solved the issue [93]. Addition of NHS forms a stable active intermediate, unlike the previous one, which then reacts with chitosan’s amino or hydroxyl group to form chitosan-polyphenol conjugate (Figure 4B). Unfortunately, the strategy of free radical synthesis to prepare chitosan-polyphenol conjugates shows low derivatization degree and hence, chemical and enzyme-mediated methods are being explored recently [94]. In case of enzyme-mediated strategy, phenolic compounds are converted to O-quinones using oxidases, a very reactive intermediate. Then, it attaches to the amine groups of chitosan by either a Michael type or Schiff base reaction demonstrated in the Figure 5A. On the contrary, a mixture of hydrogen peroxide and ascorbic acid is used to generate a redox system that induces amino, hydroxyl, and α-methylene groups of chitosan to form radicals in the free radical induced method. The polyphenols are then attached to the specific radicals through covalent bond formation (Figure 5B). The position of conjugation remains unclear and needs more understanding of the mechanisms to elucidate the bond formation between chitosan and polyphenols [86]. As mentioned in the introduction, chitosan-polyphenol conjugates overcome the limitations of both compounds when being used alone as therapeutic agents. The half-life of polyphenols in the human body is about 1–28 h which is too low to exert significant bioactive effects [95,96]. However, studies have shown that the half-life of chitosan can be extended up to 84 days by increasing the degree of deacetylation, which lowers its degradation rate [97]. Hence, it is evident and proven that the shelf-life of polyphenols can be increased by conjugation with chitosan. This prevents polyphenol degradation in biological environments [98,99].

It is also well known that both chitosan and polyphenol in their native form show antioxidant properties, but they lose their activity due to certain limitations in biological environment. In recent years, there are several studies that demonstrate that the antioxidant ability of chitosan and polyphenols can be improved by conjugation. Pasanphan and co-workers designed a chitosan-GA conjugate that imparted a synergistic scavenging ability that was greater than chitosan alone, but not significant as compared to GA [85]. Similarly, in another study the conjugate showed an increase of radical scavenging activity (92.26%) when compared to chitosan control (28%). Authors also observed increase in antioxidant activity with increase in GA content. Conjugation of GA onto chitosan enhanced the ability of GA to donate hydrogen/electron [64]. Similarly, the conjugates were also tested for their inhibitory action on the intracellular ROS in vivo. The results demonstrated that the chitosan-polyphenol conjugates were cytocompatible and able to significantly inhibit the intracellular ROS that was formed in the mouse [64]. In another study, Rui et al., have also proved the significant efficiency of chitosan-polyphenol conjugates for radical scavenging ability rather than being used alone [100].

The next section describes the efficiency of chitosan conjugated with polyphenols which include GA, Curcumin, Catechin, and Quercetin for treatment of various diseases, for which the overview has been presented in the Figure 6.

### 2.1. Chitosan-Gallic Acid Conjugates

GA also known as 3,4,5-tri hydroxy benzoic acid, is an abundant polyphenol primarily found in the bark of oak species and *Boswellia dalzielii*, an African tree [104]. It is a colorless or slightly yellow colored polyphenol with extensive pharmaceutical applications [105]. Its properties include anti-microbial [106,107], anti-obesity [108,109], and anti-oxidant properties [106,110,111]. Owing to the structural chemistry of GA, upon conjugation of GA with chitosan, it becomes easier to improve the water solubility, antioxidant properties, and metal chelating ability of the conjugate. Properties such as high reducing potential, low dissociation enthalpy of OH bonds present on the benzene ring, bulky groups obstructing the hydrogen bonds of chitosan, and hydrophilicity of GA are attractive results of being conjugated with chitosan [85]. Chemical synthesis of chitosan-GA is represented in Figure 7, which utilizes NHS/EDC for intermediate generation, and then binds to the reactive amine group of chitosan. Recently, GA has been conjugated with chitosan through various strategies imparting certain advantages and enhancing the properties of GA. Lišková et al., has enriched chitosan, sodium beta-glycerophosphate (Na-β-GP) and alkaline phosphatase (ALP)with phloroglucinol (PG) and GA to form injectable hydrogels [112]. These hydrogels were prepared for bone regeneration and their physicochemical and biological properties were characterized. The addition of polyphenols had no substantial negative effect on the hydrogel system for colony forming ability of *E. coli* in liquid medium. Antioxidant and free radical scavenging activity was observed to be higher in case of unmineralized hydrogels containing polyphenols rather than the mineralized state due to steric hindrances of concentrated minerals around GA (Figure 8). GA addition increased the antibacterial activity (Figure 8II) but was toxic and affected the cell growth. However, there is no negative effect on growth and cell adhesion of MG63 cell lines [112]. Coupling the properties of chitosan and polyphenol, Giacomo et al., developed a system that was a combination of hydroxyapatite, chitosan for modulated swelling, and release of polyphenols extracted from grape pomace. It was observed that the system remained stable at normal physiological conditions, whereas a high release profile was seen in case of inflammation. Moreover, some amounts of polyphenols were still bound to the complex after being released, so the formulation maintained longer time antioxidant and radical scavenging activities (Figure 9) [88]. In a similar study, Liu et al., grafted GA onto chitosan (GA-g-CS) by the novel process of free radical synthesis. GA-g-CS showed high inhibitory activity for α-amylase and α-glucosidase, which concludes the potential of the graft for anti-diabetic properties. However, the graft was observed to have less thermal stability and crystallinity as compared to chitosan [113].

It is reported that, the calcium oxalate crystals can be formed in the presence of chitosan which is responsible for kidney stones; hence, researchers have proved the efficiency of chitosan-GA conjugates to overcome the limitation. Queiroz et al. [114], have observed the high anti-oxidant and calcium oxalate crystals reduction ability of the conjugate. Hence, the chitosan-GA conjugate can form an efficient platform for the treatment of kidney stones in near future.

### 2.2. Chitosan-Curcumin Conjugates

Curcumin is a yellow pigment polyphenol found primarily in turmeric, naturally possessing anti-inflammatory, antioxidants and anti-cancer properties [115]. However, low solubility and the rapid metabolic nature of curcumin limit its broad applicability. Functionalized chitosan such as carboxymethyl chitosan (CMCS) has been widely used as a conjugate for increasing the bioavailability and as an efficient absorption agent for curcumin. The presence of NH_2_ and COOH active groups in CMCS is available for binding to functional groups of hydrophobic curcumin, which helps to enhance the therapeutic capability [116]. The schematic representation of chemical synthesis of chitosan-curcumin conjugate is shown in Figure 10. In another study, authors have conjugated curcumin with alginate, chitosan and pluronic F127. It was observed that pluronic F127 enhanced the solubility of curcumin. The successful delivery of the curcumin loaded composite was confirmed by inverted fluorescent microscope images of HeLa cells (green fluorescence) shown in Figure 11, and was found to be nontoxic at a concentration of 500 μg/mL. From the above experiment it was evident that the conjugate had potential anti-cancerous ability [117]. Curcumin is a lipophilic compound and hence, is susceptible to reticuloendothelial uptake. Therefore, Duan et al., have designed a novel conjugate that includes cationic poly (butyl) cyanoacrylate NPs conjugated with chitosan (PBCA) for encapsulation and increased therapeutic concentration of the delivered curcumin at its target. This system can enable parenteral means of administration of curcumin in an aqueous phase medium to utilize the efficiency of this agent against cancer. The authors tested the anti-cancer activity of the curcumin-PBCA conjugate in human hepatocellular carcinoma cells, which showed a decrease in the growth of hepatic carcinoma cells (Figure 12) and angiogenic effects [118]. In the continuation of the above study, doxorubicin (DOX) and curcumin were co-encapsulated in poly (butyl cyanoacrylate) NPs (PBCA-NPs) to access the anticancer and multi drug resistance reversal properties [119]. In a similar study, authors have encapsulated curcumin-γ-hydroxypropyl cyclodextrin (CUR-CD) hollow spheres into chitosan to form CUR-CD-CS NPs by using spray drying method. The efficacy of the formulation was tested against SCC25 cell lines, which were observed to be highly toxic when compared to the control groups. The cytotoxicity results showed 100% apoptosis in the human cancer cell line SCC25 as shown in Figure 13. Experimental results also proved that chitosan not only helps to enhance the solubility of curcumin but it also boosts its cellular uptake [120]. Researchers reported the use of curcumin NP and tripolymeric composite (chitosan, poly-g-glutamic acid, and pluronic) as a delivery agent for accelerating wound healing (Figure 14A,B). The composite was prepared using the simple technique of ionic gelation, and pluronic was used for enhancing the solubility of curcumin, lowering the bacterial infection and inflammation while wound healing. In vitro analysis confirmed the controlled release of curcumin in a simulated skin model. Neo collagen regeneration and enhanced tissue reconstruction was also observed when the conjugate was applied in an in vivo model [121].

### 2.3. Chitosan-Catechin Conjugates

Catechin is a polyphenol belonging to the flavonoid group which is known for its excellent antioxidant properties [123,124,125]. Studies, including catechin and chitosan formulations have been performed to investigate its antidiabetic potential due to the presence of phenolic groups. The interactions between chitosan-catechin are hydrogen bonding, hydrophobic interaction, and/or van der Waals contacts, and it is seen that more stable conjugates are formed with catechins [126]. Catechin-grafted chitosan (catechin-g-chitosan) shows a much higher hydroxylradical scavenging activity (46.81%) and DPPH radical scavenging activity (67.08%) than that of the control at 1mg/mL. An in vitro study of the graft for antidiabetic potential showed an increased α-glucosidase inhibitory effect and a low α-amylase inhibitory effect. The incredible antioxidant and antidiabetic potential of the graft can be a promising therapeutic tool in the biomedical sciences and pharmaceutical industry [127]. In another study, the free radical mediated conjugation method was used to conjugate chitosan-catechin and was tested for its potential activity for H_2_O_2_-induced hepatic damage in HeLa derivative human normal Chang liver cells. The conjugate was observed to reduce lipid peroxidation and intracellular ROS generation. It also increases the level of glutathione (both in normal and under OS conditions). EGCG is the most potent anti-proliferative property possessing polyphenol. EGCG significantly induces cell cycle arrest in the G1 phase and hence, induces cell apoptosis. The free radical synthesis mechanism of chitosan-catechin has been demonstrated in Figure 15. Siddiqui et al. [128] have demonstrated pro-apoptotic effects and anti-proliferative effect of EGCG against human melanoma cell growth both in vitro and in vivo. The authors had already demonstrated the above mentioned properties in their previous paper [129], but the doses were not achievable at physiological levels of EGCG. Hence, they have tried introducing a novel concept of nano chemoprevention, where the EGCG was encapsulated in chitosan NPs suitable for oral consumption that remained stable in acidic conditions. The formulation had 8-fold dose advantages over the native EGCG when tested against human melanoma Mel 928 cells. The tumor cells showed inhibition of cell cycle and induction of apoptosis in in vitro. The in vivo studies in tumorous mice, it inhibited proliferation of PCNA and Ki 67 inducing apoptosis (Figure 16). To overcome the limitation of polyphenol bio-availability, a concept of nano chemoprevention was introduced by Naghma et al. [130]. The authors have synthesized chitosan epigallocatechin-3-gallate NPs (Chit-nanoEGCG) for the slow and controlled release of polyphenol to treat prostate cancer. The EGCG was released slowly and in a faster rate in simulated gastric juice (acidic) and in simulated intestinal fluid, respectively. Chit-nanoEGCG secreted a higher prostate-specific antigen as compared to the control groups and also showed significant inactivation of tumor cells (Figure 17). The efficacy of the Chit-nanoEGCG was also investigated in tumor cells of the mouse. The observed events included induced poly (ADP-ribose) polymerases cleavage, increased Bax protein expression, decreased Bcl-2, activated caspases with reduced Ki-67, and proliferating cell nuclear antigen. They have chosen chitosan to protect the polyphenol degradation in acidic pH and to maintain its slow-release profile during oral administration. Chit-nano EGCG has been tested in preclinical setup to treat prostate cancer, their results provide a non-invasive potential platform to replace many harmful procedures like chemo and radiation therapy.

### 2.4. Chitosan-Quercetin Conjugates

Quercetin is another flavonoid with tremendous therapeutic potential, but due to its hydrophilicity and low percutaneous absorption, that limits its wide applications in vivo [131,132]. The conjugation of quercetin onto chitosan can further enhance the absorption and efficiency of quercetin in biological environments. The chitosan-quercetin conjugate through free radical mechanism, which is shown in Figure 18. In a recent study, the bioavailability and solubility of paclitaxel (PTX) is increased by carboxymethyl chitosan-quercetin (CQ) polymeric micelles. The formulation helps the drug bypassing multidrug transporter P-glycoprotein (P-gp) efflux pump. This also shows sustained-release profile as compared to the control and hence, can be a helpful agent for oral delivery of anticancer drugs that are water-insoluble [133]. Another study by Vedakumari et al. [134] studied the effect of conjugated chitosan-fibrin composite (CF) scaffolds containing quercetin for wound healing applications using a freeze-drying technique. The scaffold exhibited antibacterial effect against *Escherichia coli* (abundantly found on wound bed) and was proved to be non-toxic through MTT (3-(4,5-dimethylthiazolyl-2)-2, 5-diphenyltetrazolium bromide) assay. High mechanical strength and maximum tensile strength was also observed, which is a property of an ideal wound dressing. In vivo studies with topical application of the dressing on albino rat excision wounds showed excellent wound healing properties (Figure 19). The efficiency of quercetin-chitosan NPs has been intensively studied for sustained release in HCT 116 cell lines of colorectal cancer treatment. In vitro release profile and in vivo anti-cancer properties have proven the potential of the formulation for oral delivery of therapeutics in colorectal cancer [135]. In another study, authors prepared a pH-responsive Chitosan, Quercetin, Citraconic anhydride nanomicelle, that enhanced the inhibitory activity of quercetin against multidrug resistance (MDR) related tumor therapy when combined with the drug DOX. The nanomicelle bypassed the lysosomal degradation and also enhanced the uptake of DOX by the MCF-7/ADR cell line. This formulation provides a potential platform for the delivery of quercetin and DOX to the MDR cancer cells [136]. Rashedi et al. [137] studied the cross-linked chitosan-quercetin nanoparticles (NPs) to treat induced adenocarcinoma tumors in the colon of male Wistar rats. Histopathological studies (Figure 20) showed a high apoptosis rate, decreased microvascular density, and mitosis count that was a result of target-specific controlled release of quercetin by CS-NPs. Tzankova et al., have designed quercetin loaded chitosan/alginate NPs to treat ROS induced liver impairment in human hepatoma HepG2 cells in vitro. They have also shown decreased cell viability when the HepG2 cells were pretreated, and in vivo studies with paracetamol induced liver injury in male Wistar rats significantly lowered the high levels of serum transaminases alanine transaminase and aspartate aminotransferase, lowered the lipid peroxidation, and restored glutathione’s level. The effect of quercetin was seen to be much higher in case of encapsulation with chitosan/alginate rather than the quercetin alone. The study suggested the potential of chitosan/alginate encapsulation that can enhance the activity of quercetin to treat stress induced injuries [138].

## 3. Conclusions and Future Prospective

In recent years, natural antioxidants have been in great demand due to their potential therapeutic advantages for human health. Both chitosan and polyphenols are well-recognized antioxidant and antimicrobial agents, but both suffer due to certain inherent limitations, which reduce their therapeutic efficacy in the biological environment. Hence, the conjugation of chitosan-polyphenol emerges as a potential candidate to overcome the drawbacks of both compounds and impart synergistic health benefits over native ones. Though many strategies have been adopted to design chitosan-polyphenol conjugates, free radical induced conjugation was the most effective. The therapeutic efficacy of chitosan-polyphenols is attributed to the presence of an active amine group in chitosan and phenolic hydroxyl groups of polyphenols. Several studies have been conducted to evaluate the chitosan-polyphenol therapeutic efficacy that can be used against various deadly diseases like infectious, neurodegenerative, cancer, diabetes, and CVD, etc. Although in vitro analyses yield much success, their applicability in human trials in clinic is still limited. Some studies have observed controversial results, such as sub-chronic and oral toxicity in animal studies. Therefore, detailed studies need to be performed to evaluate the safety and efficacy on human health. Extensive research on optimal safe doses and modes of administration is needed to be evaluated to avoid the toxic effects of the chitosan-polyphenols conjugate. Determining the action of specific polyphenols against a particular disease is still a significant concern to be investigated in human health. Hence, developing reliable methods to measure the individual polyphenol activity in body tissues and fluids with their physiological relevance to a particular disease is highly necessary to target the exact pathway with efficient polyphenolic compounds.Moreover, their applications in pharmaceutics and biomedicine are scarce due to various drawbacks of both chitosan and polyphenols, mainly being less stable in the human body. Therefore, it is essential to understand the biosynthesis pathways by using high throughput techniques. Improvements in the conjugation methods and nanotechnology can improve their in vivo applications in the future, widening the scope of pharmaceutical companies and biomedical fields. There is a lot of future research that needs to be done, but chitosan-polyphenol conjugates undoubtedly offer great potential for human health.

## Figures and Tables

**Figure 1 life-12-01768-f001:**
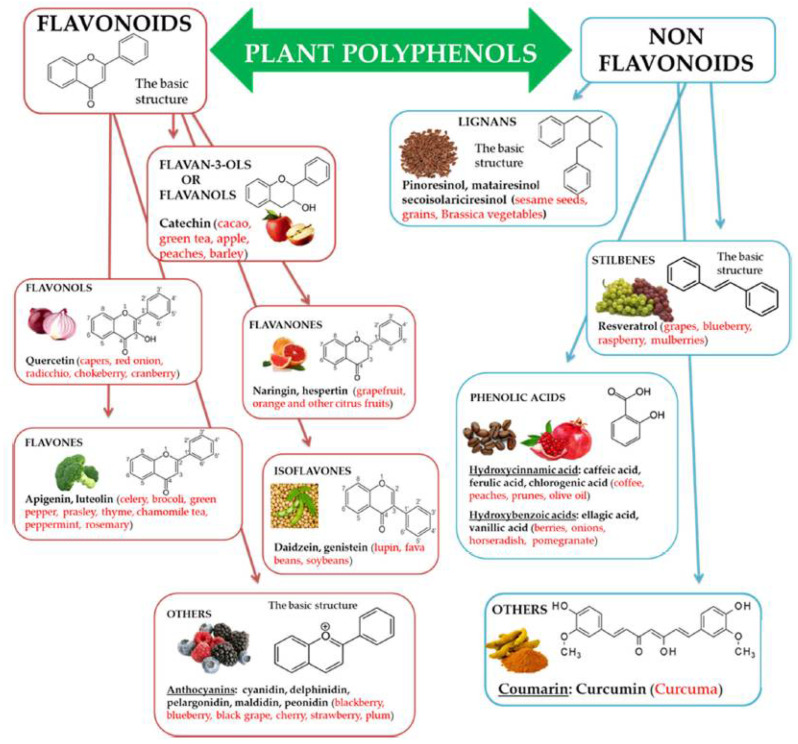
Pictorial representation of different classes and subclasses of polyphenols, including their chemical structures and sources, adapted from Ref. [11].

**Figure 2 life-12-01768-f002:**
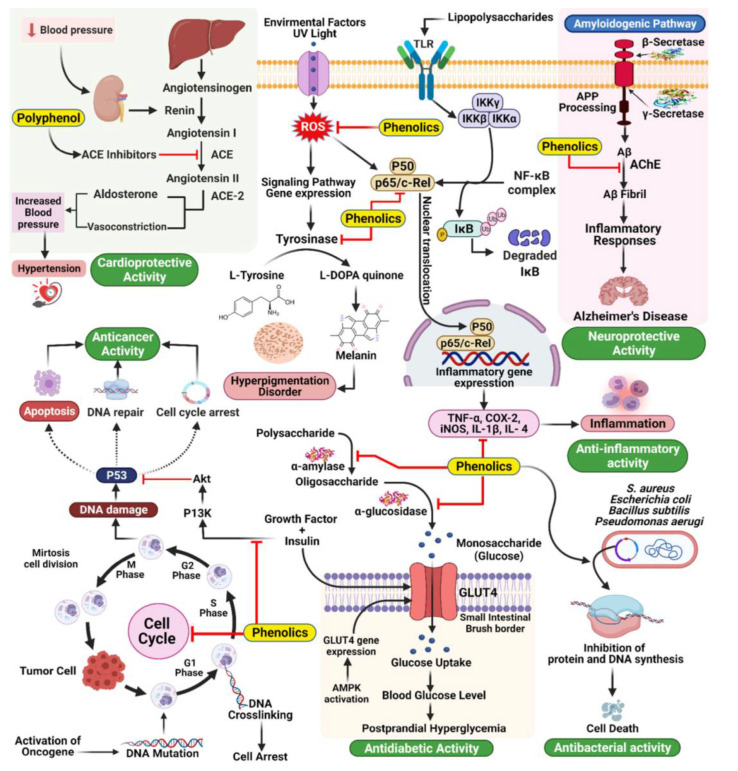
Mechanism of action underlying the therapeutic activities of phenolics in different diseases, adapted from Ref. [52].

**Figure 3 life-12-01768-f003:**
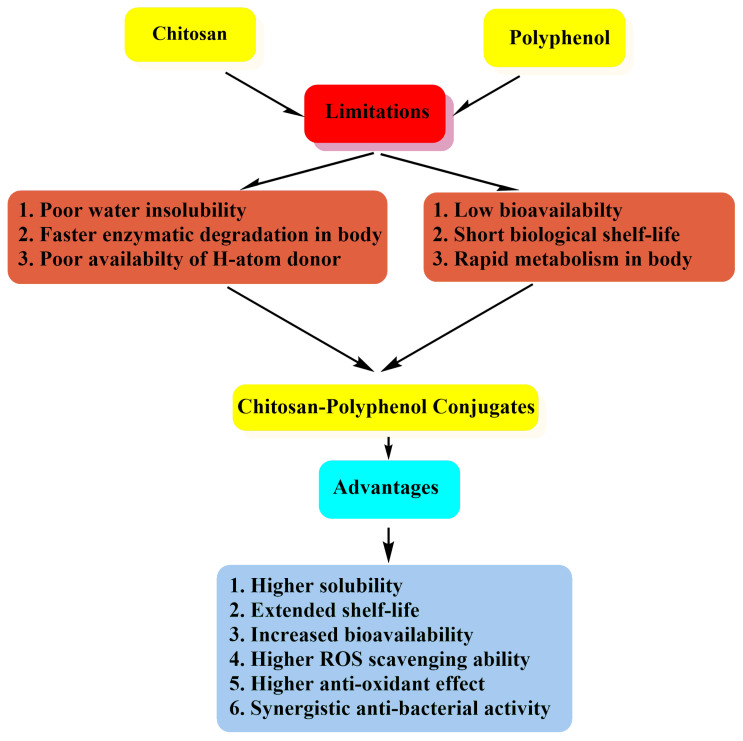
Flow diagram of limitations of chitosan and polyphenol, and advantages of chitosan-polyphenol.

**Figure 4 life-12-01768-f004:**
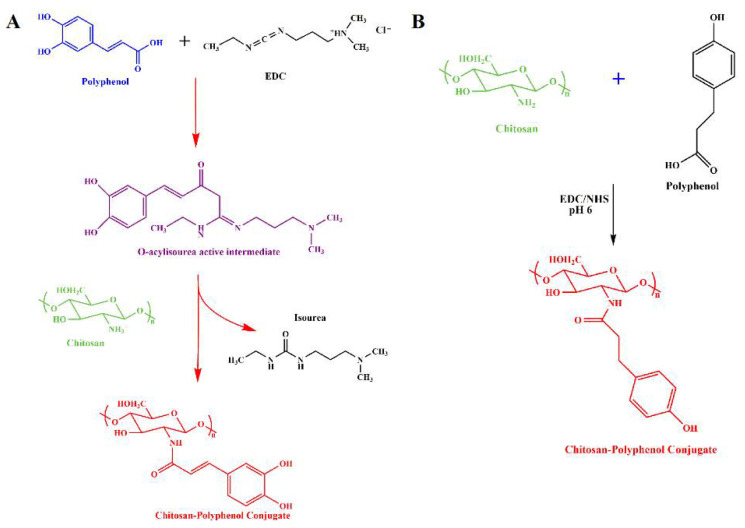
Proposed mechanism for chemical synthesis of polyphenol-chitosan conjugate (**A**) using only EDC, adapted with permission from Ref. [101], © 2022 The Society for Biotechnology, Japan, Elsevier B.V. (**B**) using EDC/NHS, adapted with permission from Ref. [102], © 2022 The Royal Society of Chemistry.

**Figure 5 life-12-01768-f005:**
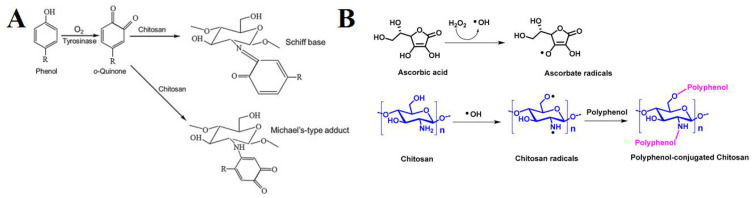
Proposed mechanism for polyphenol-chitosan conjugate through (**A**) enzyme-mediated strategy, adapted with permission from Ref. [103], © 2022 Elsevier B.V., and (**B**) free radical induced reaction, adapted from Ref. [93].

**Figure 6 life-12-01768-f006:**
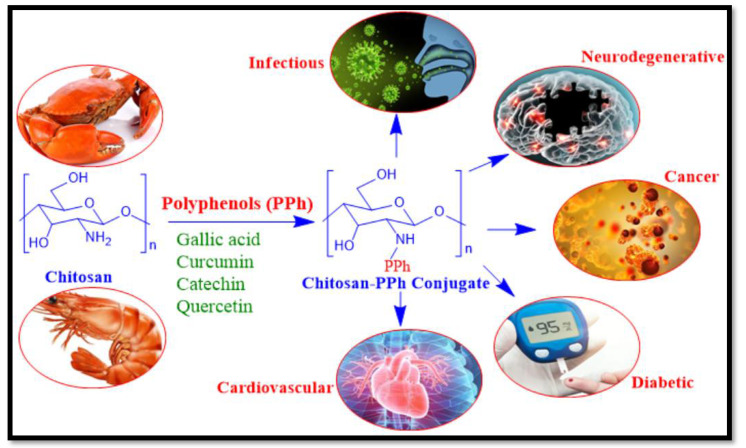
Overview of chitosan-polyphenol conjugates for treatment of various diseases.

**Figure 7 life-12-01768-f007:**
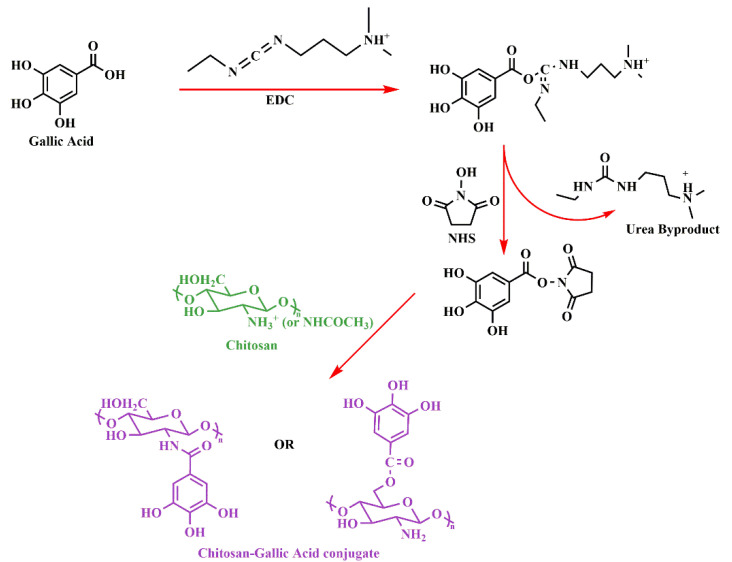
Ester modification synthesis of chitosan-GA conjugation.

**Figure 8 life-12-01768-f008:**
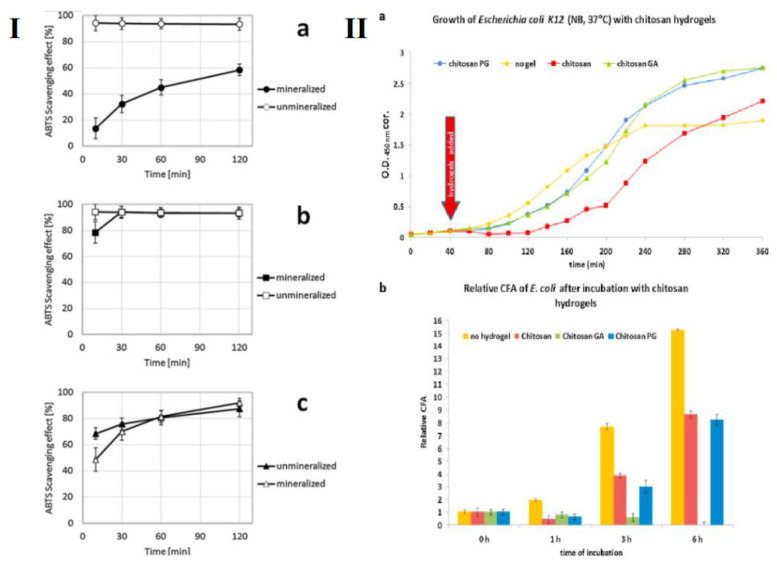
(**I**) Antioxidant activity of chitosan hydrogels enriched with GA (**a**), PG (**b**), and without tannins (**c**) before and after mineralization. (**II**) Antibacterial activity of chitosan hydrogels enriched with GA and PG. (**a**) Growth of E. coli in liquid culture in presence of hydrogels (**b**) Relative colony-forming ability of E. coli on agar after incubation with hydrogels (CFU counts were normalized to values at time 0 h). Mean values are shown (*n* = 4). Error bars show standard deviation, adapted with permission from Ref. [112], © 2022 Elsevier Ltd.

**Figure 9 life-12-01768-f009:**
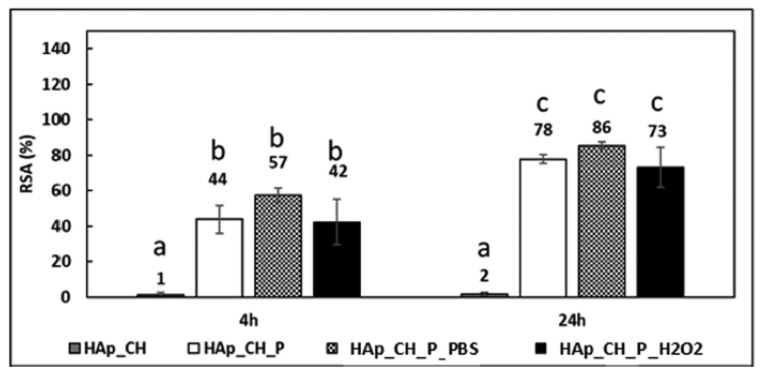
Radical Scavenging Activity obtained with the DPPH test, the values are expressed in percentage, and they were evaluated at 4 and 24 h of soaking in the DPPH solution. The bars that share at least one letter are not significantly different (*p* < 0.05 calculated with Tukey’s test), adapted with permission from Ref. [88], © 2022 American Chemical Society.

**Figure 10 life-12-01768-f010:**
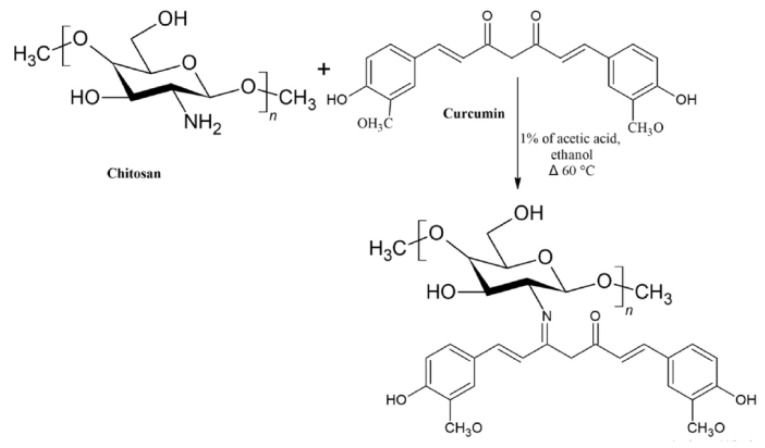
Chemical synthesis of chitosan-curcumin conjugation, adapted with permission from Ref. [122], © 2022 Elsevier Ltd.

**Figure 11 life-12-01768-f011:**
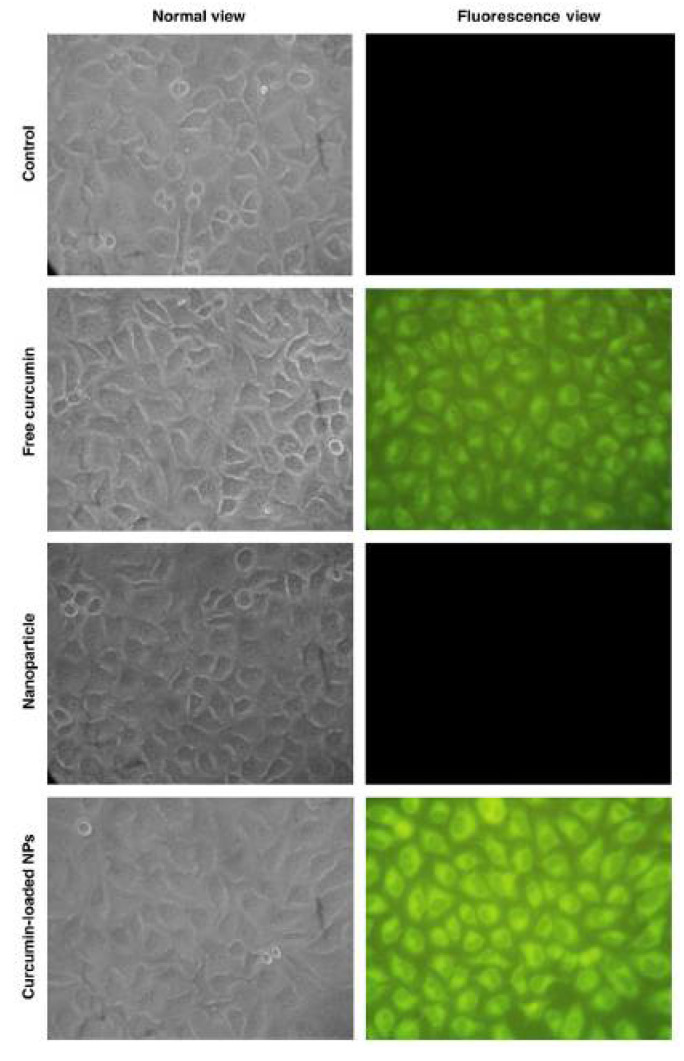
Images of HeLa cells as visualized under inverted fluorescent microscope, adapted with permission from Ref. [117], © 2022 Elsevier Inc.

**Figure 12 life-12-01768-f012:**
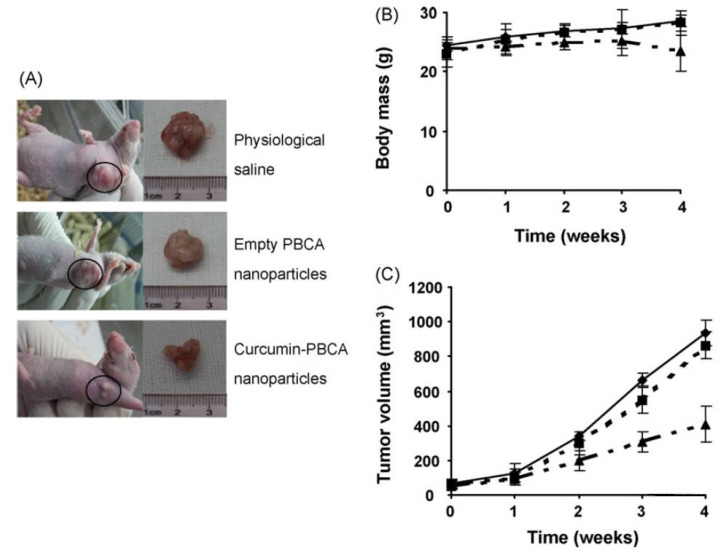
In vivo efficacy of curcumin-PBCA NPs and empty PBCA NPs in HepG2 hepatocellular cancer xenograft. (**A**) The images of HepG2 xenograft-bearing mice treated with curcumin-PBCA NPs, empty PBCA NPs and control tumor; changes in body weight of animals (**B**) and tumor volume (**C**) as a function of time in subcutaneous HepG2 xenograft. Straight line: physiological saline; dashed line: empty PBCA NPs; double dot dashed line: curcumin PBCA NPs, adapted with permission from Ref. [118], © 2022 Elsevier B.V.

**Figure 13 life-12-01768-f013:**
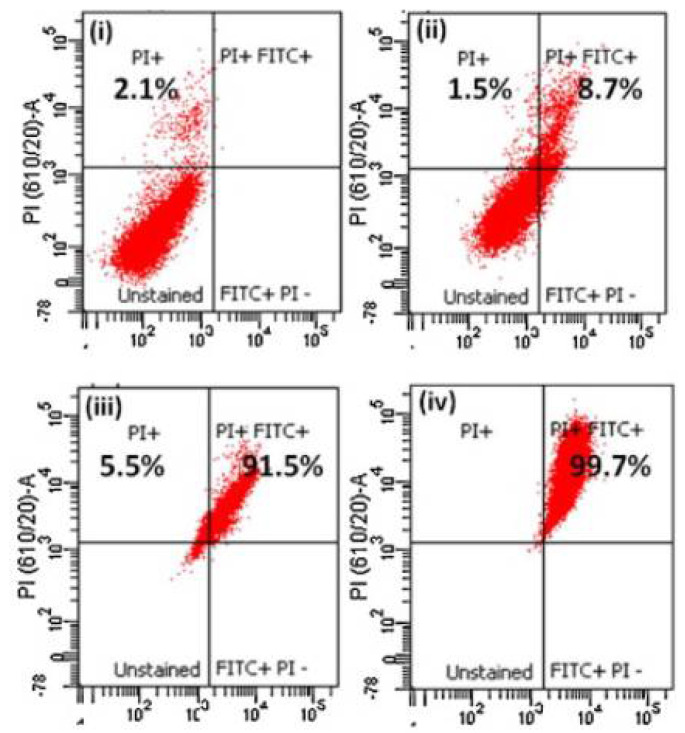
% Apoptosis of SCC25 cells by CUR determined by Annexin V-FITC/PI staining and flow cytometry (**i**) control cells, (**ii**) CUR, (**iii**) CUR-CS, (**iv**) CUR-CD-CS, adapted with permission from Ref. [120], © 2022 Elsevier B.V.

**Figure 14 life-12-01768-f014:**
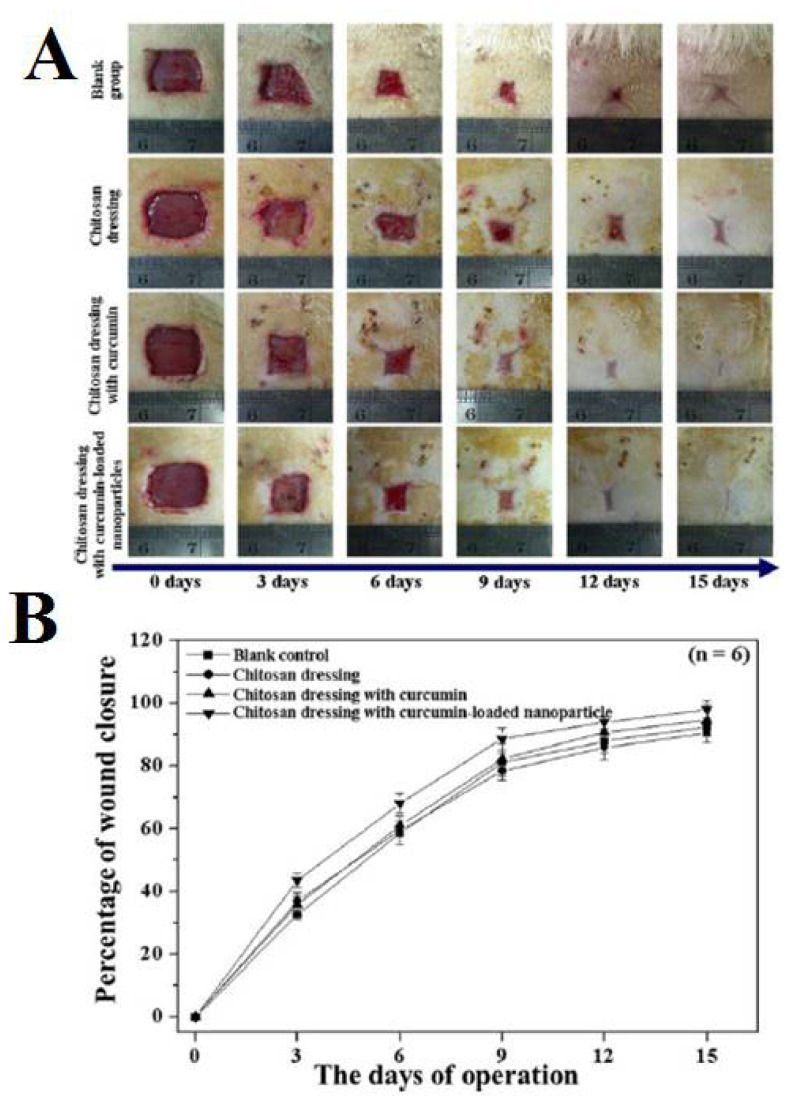
(**A**) Representative photographs of wounds following full thickness skin excision at 3, 6, 9, 12, and 15 days after surgery; (**B**) Percentage wound closure is presented at the indicated time points, adapted with permission from Ref. [121], © 2022 Wiley Periodicals, Inc. J Biomed Mater Res Part B: Appl Biomater.

**Figure 15 life-12-01768-f015:**
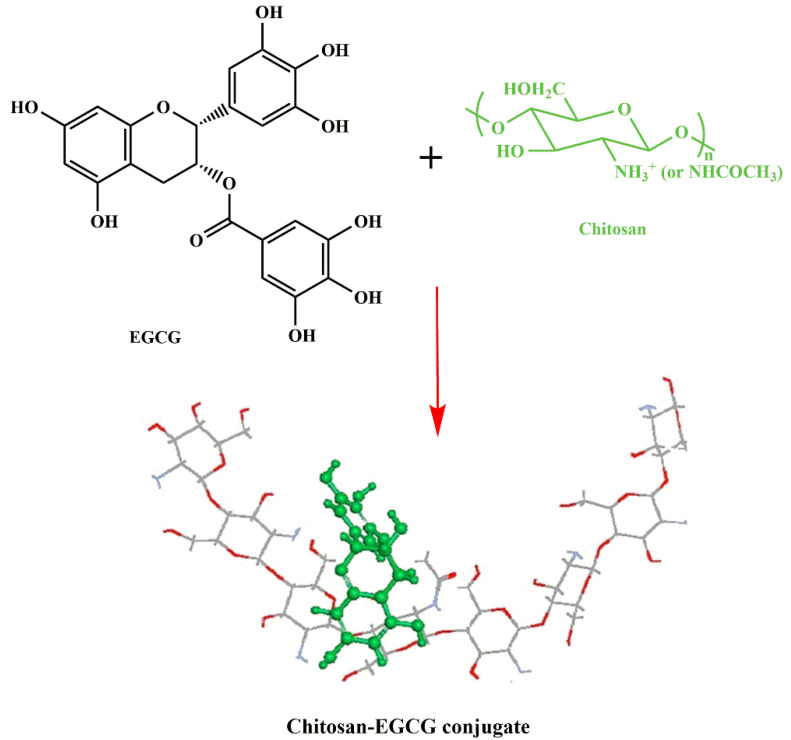
Free radical mechanism of chitosan-catechin conjugation, adapted with permission from Ref. [126], © 2022 Elsevier Ltd.

**Figure 16 life-12-01768-f016:**
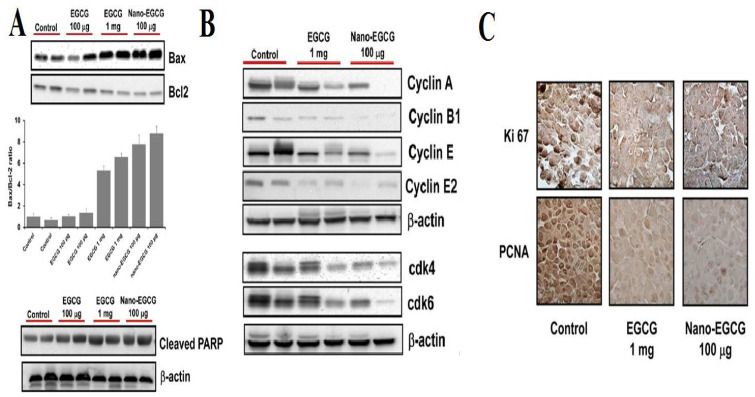
Comparative effects of nonencapsulated EGCG and nano-EGCG on markers of apoptosis, cell cycle, and Proliferation markers in tumors isolated from nude mice. (**A**) Protein expression of Bax, Bcl2, and the Bax/Bcl2 ratio and PARP. (**B**) Protein expression of Cyclins A, B1, E, E2, CDK 4, and CDK6. The cells were treated with each agent and harvested 24 h after treatment. Equal loading was confirmed by stripping the membrane and reprobing it with β-actin. Each experiment was repeated twice with similar results. (**C**) Effect of the treatments on expression of Ki-67 and PCNA in tumor tissues isolated from athymic nude mice. Tumor sections were stained using specific antibodies as detailed in Materials and Methods. Counterstaining was performed with hematoxylin. Scale bar, 50 μm. Photomicrographs (magnification, 20×) show representative pictures from two independent samples, adapted with permission from Ref. [128], © 2022 Elsevier Inc.

**Figure 17 life-12-01768-f017:**
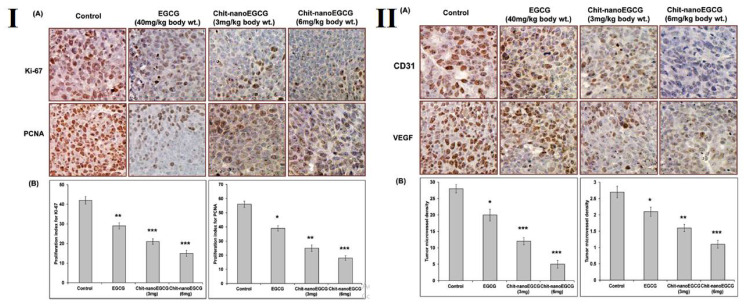
(**I**) (**A**) Effect of Chit-nanoEGCG on expression of Ki-67 and PCNA in tumor tissues of athymic nude mice. Tumor sections from athymic nude mice were stained using specific antibodies as detailed in Materials and Methods. Counterstaining was performed with hematoxylin. Scale bar, 50 μm. Photomicrographs (magnification, ×20) show representative pictures from two independent samples. (**B**) Proliferation index for Ki-67 (left panel) and PCNA (right panel) is shown. * *p* < 0.05 and ** *p* < 0.01 and *** *p* < 0.001, versus control group. (**II**) (**A**) Effect of Chit-nanoEGCG on expression of CD31 and VEGF in tumor tissues of athymic nude mice. Tumor sections from athymic nude mice were stained using specific antibodies as detailed in Materials and Methods. Counterstaining was performed with hematoxylin. Scale bar, 50 μm. Photomicrographs (magnification, 20×) show representative pictures from two independent samples. (**B**) Tumor microvessel density (left panel) and VEGF immunoreactivity score (right panel) was scored as 0+ (no staining), 1+ (weak staining), 2+ (moderate staining), 3+ (strong staining), and 4+ (very strong staining). * *p* < 0.05 and ** *p* < 0.01 and *** *p* < 0.001, versus control group, adapted with permission from Ref. [130], © 2022 Oxford University Press.

**Figure 18 life-12-01768-f018:**
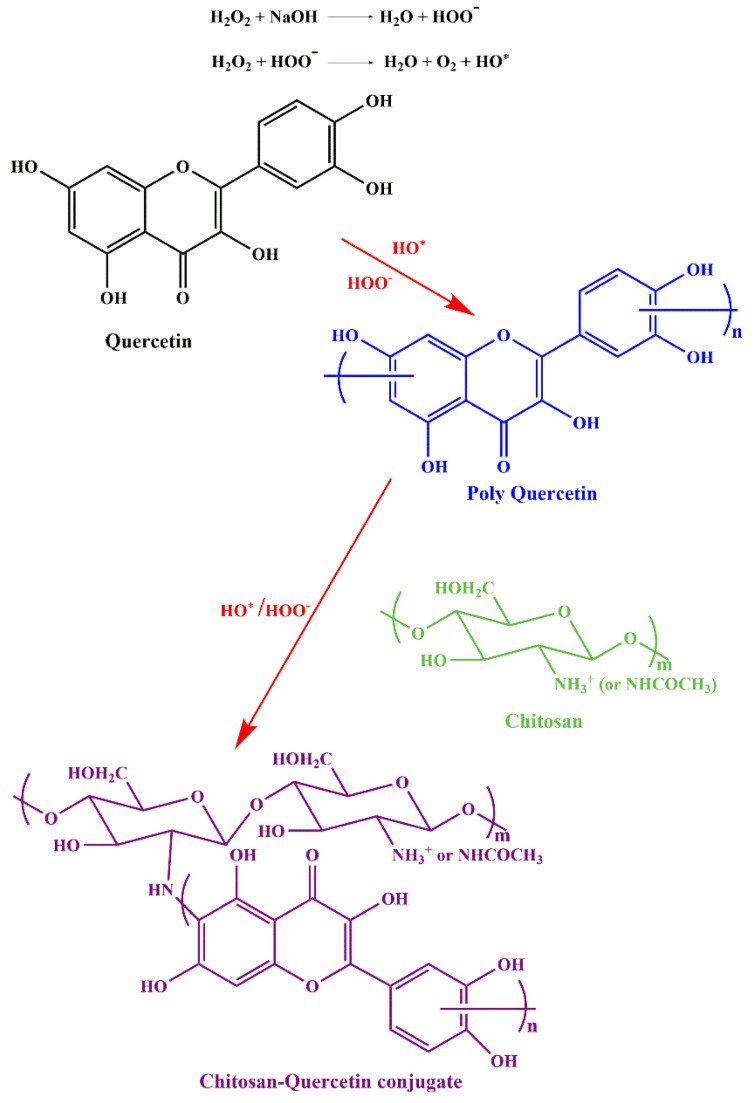
Synthesis of chitosan-quercetin conjugate by free radical strategy, adapted with permission from Ref. [94], © 2022 WILEY-VCH Verlag GmbH & Co. KGaA, Weinheim.

**Figure 19 life-12-01768-f019:**
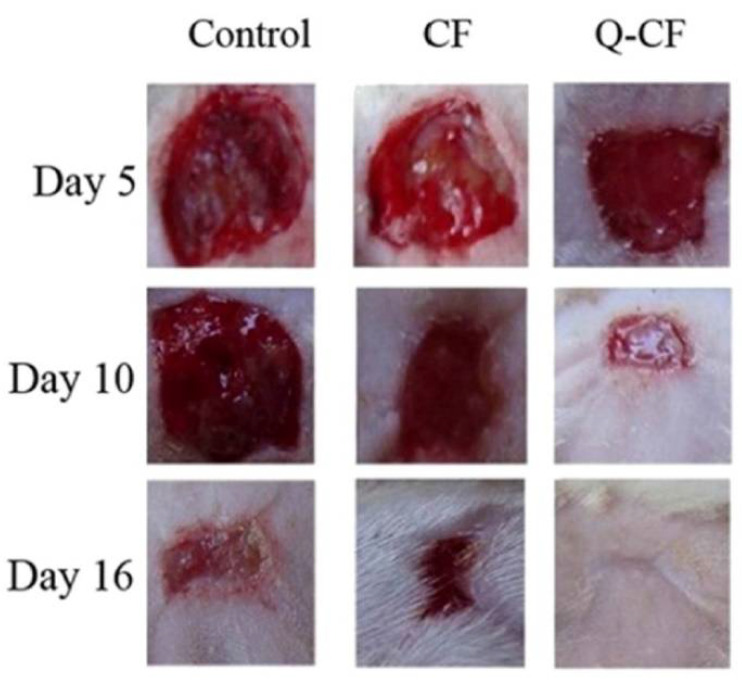
Photographic evaluation of wound closure on different days of healing in control and CF and Q-CF treated rats, adapted with permission from Ref. [134], © 2022 Elsevier B.V.

**Figure 20 life-12-01768-f020:**
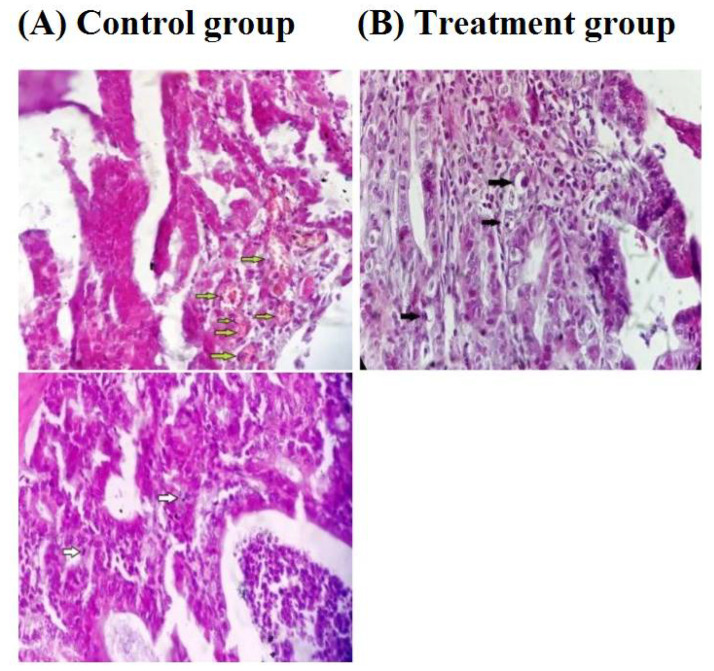
Histological sections of adenocarcinomas from Wistar rat colon. Paraffin embedded sections were stained with H&E. (**A**) Cancerous control group, (**B**) Treatment group; cancerous rats were administered by Qu loaded CS NPs through enema. The black arrows indicate apoptotic bodies and the white and green arrows indicate mitotic cells and microvasculars, respectively (400×) Adapted from Ref. [137].

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
