# Peer review of "Chitosan-Polyphenol Conjugates for Human Health"

_life, 2022, doi:10.3390/life12111768_

Round 1

Reviewer 1 Report

1. Abstract: The significance of the direction of chitosan-polyphenol complexes is not deep enough and needs to be extended to improve the innovation of this paper.

2. The therapeutic application of polyphenols has not been determined. The author needs to provide enough evidence or modify the relevant description.

3. Figure 6: It is recommended to redraw the relevant chemical structure yourself.

4. Figure 15 and 17: Remade, the quality of these figures needs to improve.

5. Figure 19 was not sufficient or new for this paper; replacement or deletion recommended

6. Line 380-389: Wordy and irrelevant to the conclusion. It is recommended to use 1-2 sentences to summarize the content.

7. Line 393-395: What does the author say about this principle in the text? If not, it is recommended to add relevant content.

8. Insufficient Perspectives. Adding prospect content or modifying the title of this section is recommended.

Author Response

Answer to the Reviewers Comments

 Dear Reviewer,

Thank you so much for your time to go through our review manuscript and provide suggestions and comments to enhance the quality of the manuscript. As per your kind instructions, the authors have tried their best to implement all your suggestions and comments.

 Reviewer Comments

  1. Abstract: The significance of the direction of chitosan-polyphenol complexes is not deep enough and needs to be extended to improve the innovation of this paper.

Answer: Thank you so much for your suggestion; we have included (with track change) the significant direction of chitosan-polyphenol conjugates in the abstract.

  1. The therapeutic application of polyphenols has not been determined. The author needs to provide enough evidence or modify the relevant description.

Answer: As per your instruction, the therapeutic applications of polyphenols have been explained with references. 

  1. Figure 6: It is recommended to redraw the relevant chemical structure yourself.

Answer: The figure has been remade in ChemDraw and replaced.

  1. Figure 15 and 17: Remade, the quality of these figures needs to improve.

Answer: Thank you so much for the suggestion, authors completely agree with your concern, but the figure was taken from a research article, hence it is difficult to make further modifications in Figure 15. Authors hope you will understand our limitations. As per your instruction, figure 17 has been remade in ChemDraw and replaced.

  1. Figure 19 was not sufficient or new for this paper; replacement or deletion recommended

Answer: As per your instruction, Figure 19 has been replaced.

  1. Line 380-389: Wordy and irrelevant to the conclusion. It is recommended to use 1-2 sentences to summarize the content.

Answer: Thank you so much for the suggestion. The conclusion has been rewritten.

  1. Line 393-395: What does the author say about this principle in the text? If not, it is recommended to add relevant content.

Answer: Thank you so much for the suggestion. The conclusion has been rewritten.

  1. Insufficient Perspectives. Adding prospect content or modifying the title of this section is recommended.

Answer: Thank you so much for the suggestion, future perspective has been added in the section.

Reviewer 2 Report

Abstract: Please elaborate the biochemical functions at cellular/subcellular level wit chitosan-polyphenols conjugate over either of the two alone. Do we observe a higher biological half life with such conjugations ?.

Introduction : Please provide the bibliometric analysis of different issues of chitosan-polyphenols conjugation in human health. Please ensure that no original figure is taken from any previous publications or else due permission is sought after from the authors( Fig. 1). Can we enlist adavantages and limitations of chitosan-polyphenols conjugation through a Flow Diagram ?. ( Line 85-86). 

2. Chitosan-polyphenols conjugates : Likewise Fig 3 and /fig. 4 , please ensure , it is not  a copyright infringement ( line 124-129). Authors should analyse such conjugates like structural manifestations,, physiological/biochemical manifestations, shelf life of conjugation,  free radicals scavenging  ability ,etc...and so on . Fig. 5 is actually not needed , can put in text form. Likewise section 2.1 , 2.2, 2.3 , 2.4 should be organised with extra care about these figures ( numbers are exceptionally high , just taken from previous publications , Fig. 6-Fig.19 )  which are simply pasted here. . These are all copyright materials , possibly involve violation. These sections above all , are written like a thesis work, instead of providing a metanalysis of the issue , therefore lacking an analytical view.  

Conclusion : What do you mean 9 line 389-391). This section ( line 391-193) should have gone under section 2.1

I am personally not convinced with this review written in such a haste . Authors should provide their own metanalysis in a synthesized manner. I , therefore , recommend the manscucript REJECT in current form . 

Author Response

Answer to the Reviewer's Comments

Dear Reviewer,

Thank you so much for your time to go through our review manuscript and provide suggestions and comments to enhance the quality of the manuscript. As per your kind instructions, the authors have tried their best to implement all your suggestions and comments.

Reviewer 

  1. Comments: Abstract: Please elaborate the biochemical functions at cellular/subcellular level wit chitosan-polyphenols conjugate over either of the two alone. Do we observe a higher biological half life with such conjugations ?.

Answer: Thank you so much for your kind suggestion; we have included the biochemical functions at the cellular/subcellular level with chitosan-polyphenols conjugates in the introduction. The biological half-life of polyphenols is around 1-28h, however, the half-life of chitosan molecules depending on their degree of deacetylation and molecular weight the half-life varies up to 84 days. Hence, it is obvious that the chitosan-polyphenol conjugates will show a higher biological half-life.

  1. Introduction : Please provide the bibliometric analysis of different issues of chitosan-polyphenols conjugation in human health. Please ensure that no original figure is taken from any previous publications or else due permission is sought after from the authors( Fig. 1). Can we enlist adavantages and limitations of chitosan-polyphenols conjugation through a Flow Diagram ?. ( Line 85-86).

Answer: Thank you so much for your valuable suggestion; we tried to incorporate the bibliometric analysis of different issues of chitosan-polyphenols conjugation via PubMed, Web of Science but unfortunately the data we observed were not significantly high enough to be plotted in a graphical manner. Regrading the original figures, we appreciate your concern about copyright, but we have already received the copyright permission from the publishers and submitted to the editor of the journal. Flow Diagram, already included in the text as per your kind instruction.

  1. Chitosan-polyphenols conjugates : Likewise Fig 3 and /fig. 4 , please ensure , it is not  a copyright infringement ( line 124-129). Authors should analyse such conjugates like structural manifestations,, physiological/biochemical manifestations, shelf life of conjugation,  free radicals scavenging  ability ,etc...and so on . Fig. 5 is actually not needed , can put in text form. Likewise section 2.1 , 2.2, 2.3 , 2.4 should be organised with extra care about these figures ( numbers are exceptionally high , just taken from previous publications , Fig. 6-Fig.19 )  which are simply pasted here. . These are all copyright materials , possibly involve violation. These sections above all , are written like a thesis work, instead of providing a metanalysis of the issue , therefore lacking an analytical view.  

Answer: Thank you so much for your suggestion; We have already received the copyright permission from the publishers and submitted it to the editor of the journal. We have added a paragraph in chitosan-polyphenol conjugates section. We understand your concern, but to describe the contents we have included the figures from the original articles and asked the publishers for their copyright permission. We have tried our best to provide sufficient details to understand the importance of the conjugates compared to their individual therapeutic efficacy in human health.

  1. Conclusion : What do you mean 9 line 389-391). This section ( line 391-193) should have gone under section 2.1

Answer: Thank you so much for your suggestion, authors have rewritten the conclusion and future perspective section.

Reviewer 3 Report

Experiments were well performed and the theme is moderately novel and very interesting, but some minor improvements are needed. Overall, the manuscript will meet the publishing standard of the journal after minor revisions.

Line 115: ‘figure 4A’ and line 123 ‘Figure 5’, It should be consistent.

Line 133: Is the first line of the paragraph formatted correctly? The later paragraphs also have this problem.

Figure 7: I suggest bar graphs with significant differences.

Figure 12: Please check the diagram's label and caption information carefully.

Author Response

Answer to the Reviewer's Comments

Dear Reviewer,

Thank you so much for your time to go through our review manuscript and provide suggestions and comments to enhance the quality of the manuscript. As per your kind instructions, the authors have tried their best to implement all your suggestions and comments.

Reviewer 

Experiments were well performed, and the theme is moderately novel and very interesting, but some minor improvements are needed. Overall, the manuscript will meet the publishing standard of the journal after minor revisions.

Comments: Line 115: ‘figure 4A’ and line 123 ‘Figure 5’, It should be consistent.

Answer: Thank you so much for the suggestion, every “Figure” in the manuscript has been changed and is consistent.

Comments: Line 133: Is the first line of the paragraph formatted correctly? The later paragraphs also have this problem.

Answer: Thank you so much for the suggestion, all the paragraphs have been formatted as per your suggestion.

Comments: Figure 7: I suggest bar graphs with significant differences.

Answer: Thank you so much for the suggestion, authors completely agree about your concern, but the figure was taken from the research article, hence it is difficult to make further modifications in the bar graph, we hope you will understand our limitation.

Comments: Figure 12: Please check the diagram's label and caption information carefully.

Answer: Authors have gone through the manuscript and checked the diagram's label and caption information carefully, and the required changes have been made.

Round 2

Reviewer 2 Report

Recommended for acceptance